# Why Are Some Male Alcohol Misuse Disorder Patients High Utilisers of Emergency Health Services? An Asian Qualitative Study

**DOI:** 10.3390/ijerph191710795

**Published:** 2022-08-30

**Authors:** Pamela Goh, Lina Amirah Binte Md Amir Ali, Donovan Ou Yong, Gabriel Ong, Jane Quek, Halitha Banu, Jun Tian Wu, Charles Chia Meng Mak, Desmond Renhao Mao

**Affiliations:** 1Home Team Behavioural Sciences Centre, Ministry of Home Affairs, Singapore 698928, Singapore; 2Division of Advanced Internal Medicine, National University Hospital, Singapore 119074, Singapore; 3Emergency Department, Tan Tock Seng Hospital, Singapore 308433, Singapore; 4Yong Loo Lin School of Medicine, National University of Singapore, Singapore 117456, Singapore; 5National Addictions Management Service, Institute of Mental Health, Singapore 539747, Singapore; 6Acute & Emergency Care Department, Khoo Teck Puat Hospital, Singapore 768828, Singapore

**Keywords:** alcohol misuse, emergency services, emergency department, repeated attendances, high utilization

## Abstract

Background: Certain alcohol misuse patients heavily utilise the Emergency Department (ED) and Emergency Medical Services (EMS) and may present with intoxication or long-term sequelae of alcohol misuse. Our study explored reasons for repeated ED/EMS utilisation and sought to understand perpetuating and protective factors for drinking. Methods: Face-to-face semi-structured qualitative interviews were conducted. Participants were recruited from an ED in Singapore. Interviews were audio-recorded, transcribed verbatim and underwent manual thematic analysis. Emergent themes were independently reviewed for agreement. Data from medical records, interview transcripts, and field notes were triangulated for analysis. Results: All participants were male (*n* = 20) with an average age of 55.6 years (*SD* = 8.86). Most were unemployed (75%), did not have tertiary education (75%), were divorced (55%), and had pre-existing psychiatric conditions (60%) and chronic cardiovascular conditions (75%). Reasons for utilisation included a perceived need due to symptoms, although sometimes it was bystanders who called the ambulance. ED/EMS was preferred due to the perceived higher quality and speed of care. Persistent drinking was attributed to social and environmental factors, and as a coping mechanism for stressors. Rehabilitation programs and meaningful activities reduced drinking tendencies. Conclusion: ED/EMS provide sought-after services for alcohol misuse patients, resulting in high utilisation. Social and medical intervention could improve drinking behaviours and decrease overall ED/EMS utilisation.

## 1. Background

Patients who frequently attend the emergency department (ED) for alcohol-related problems are known as alcohol-related frequent attenders [1] (ARFAs). While these patients account for a small percentage of the patient population treated by the ED, they utilise a disproportionately large amount of emergency medical resources, which encompass emergency medical services (EMS), ED, and inpatient services [2,3]. Compared to the average ED patient, ARFAs have lower admission and higher abscondment rates from the ED, suggesting that their visits could be inappropriate and less urgent in nature [4]. In relation to stigmatisation, emergency providers are known to question the severity or validity of their medical complaints [5,6]. Investigative studies [3,7] have shined a light on the high mortality and morbidity of this patient population.

ARFAs present to EDs with acute alcohol intoxication as well as long-term sequelae of alcohol misuse, including alcohol withdrawal symptoms, gastrointestinal bleeding, and complications of liver failure. In a time-limited setting with competing priorities, ED physicians have found it challenging to provide sustainable and meaningful long-term health interventions to reduce alcohol misuse [8].

Most ARFAs tend to default their treatment regimen for addiction [9], especially within the first three months. Local data suggest that the dropout rate could be as high as 75% in the same period. This specific patient group would turn to the ED and EMS instead, eventually resulting in unplanned hospital admissions for alcohol-related consequences.

Previous research [6,10] largely focused on the perspectives of emergency medical professionals who encountered these ARFAs in their course of work, but few [11] have examined the outlooks of ARFAs. There are limited data regarding the reasons for ARFAs’ motivations to drink and utilise emergency healthcare services. While these predominantly non-Asian studies [12,13,14] have identified epidemiological risk factors for frequent attendance at the ED, findings may not fully be generalisable to ARFAs since they were not specifically derived from the alcohol misuse population. Studies [15,16] have shown that while the prevalence of drinkers in Asia has remained stable over the last two decades, there is a significant increase in alcohol consumed per drinker. ARFAs are a group with heavy alcohol consumption driving this phenomenon. Drinking habits, beliefs, behaviours, and responses to traditional alcohol interventions are contextual. ARFAs remain a poorly understood population and may offer insight into the reasons for the intensifying alcohol consumption in Asia. 

This qualitative study aims to understand the rationale for ARFAs’ repeated use of EMS and ED. ARFAs’ alcohol drinking patterns will be explored, including reasons for drinking, and underlying perpetuating and protective factors fundamental to their overall utilisation of healthcare services. The scarcity of research on ARFAs underscores the need for a more comprehensive understanding of this group of people. This research will guide initiatives targeted at problematic drinking behaviours in these individuals, which in turn can reduce their reliance on EMS and EDs stemming from alcohol consumption.

## 2. Methodology

### 2.1. Study Design and Setting

This is a prospective, single-centre qualitative study of alcohol-misuse patients who have frequent attendance in the ED. Participants were recruited via convenience sampling from the ED of Khoo Teck Puat Hospital (KTPH). KTPH is a 795-bed general and acute care hospital serving a population of over 550,000 in Singapore’s northern region. Since its operation in 2010, KTPH’s ED has seen over 138,000 patients yearly, attending to both walk-ins as well as the more critically ill and trauma cases.

A team of trained male and female researchers conducted face-to-face interviews with study participants from February to July 2021 (6 months) (including the first, fifth, and sixth authors of this publication). Based on existing qualitative, grounded theory studies, an estimated but also practical sample size of 15 to 30 was determined. Study enrolment was pursued until theoretical saturation was achieved, with no new themes emerging from the interviews. 

### 2.2. Inclusion and Exclusion Criteria

Eligible participants (male and female) were over 21 years old and spoke either English or Mandarin. They had at least five ED visits in the preceding 12 months of the study enrolment date, of which at least two visits were alcohol-related as defined by SNOMED-CT codes (Appendix A). Patients who were psychologically or physically unable to demonstrate a capacity for informed consent or refused participation were excluded from this study. 

### 2.3. Recruitment Process

Participants were first identified by attending medical staff within the ED, before they were referred to the study team. Once the study team assessed a patient to be eligible, the study coordinator then approached them to explain the purpose of the study and that their participation entailed interviews on their chronic alcoholism and the subsequent reasons, including their experiences, for their frequent ED visits. Written informed consent was obtained only when the patient was finally deemed to be medically fit. This consent process was also maintained for patients already admitted to the inpatient wards from the ED, in which case the admitted patient was approached and recruited in the ward. Only when informed consent was obtained were the interviewers introduced to the study participants by the study coordinator. Interviewers introduced themselves as researchers who were interested in learning about their drinking behaviours and ED/EMS usage experiences. 

### 2.4. Data Collection

Each participant was interviewed by two primary interviewers with experience in qualitative research methods. A similar research area of interest that the interviewers have previously looked at encompassed understanding citizens’ involvement in responding to public medical cases before first responders arrive.

Interviews were conducted in a private room within the hospital grounds either on the same or the next day of the participant’s ED visit or on a different day convenient for both the participant and interviewers. No one else was present in the interview except for the two interviewers and the participant; all interviews were also completed on the same day within that single session. The interviews typically occurred in the morning when participants were less likely to drink and/or experience alcohol withdrawal symptoms prior to the interview. A semi-structured questionnaire was employed, covering participants’ socio-demographic background, personal alcoholism profile, social alcoholism profile, perceptions of utilising emergency services, and motivation behind corrective behaviours, as well as attempts and attitudes towards interventions (Appendix A). Interviewers were conducted until saturation was achieved, with no new themes emerging from the interviews. All interviews were audio-recorded with the verbal consent of participants.

Patient identifiers were removed, and interviews were transcribed verbatim on Microsoft Word by one of the two interviewers from each interview session. Each participant was assigned a unique subject code prior to the interview to be used throughout the audio recording. To capture essential nuances in communication, interviewers also recorded independent field notes beyond the questionnaire, including non-verbal behaviours of participants not captured by the audio recording. All audio recordings and transcripts were then stored on a password-protected laptop at the study site, accessible only by the research team.

Participants’ medical records were retrieved by their treating doctor in the ED during their medical treatment. The treating doctor would provide information on each participant’s number of ED attendances and number of alcohol-related ED attendances in the preceding 12 months, as well as any history of chronic and psychiatric conditions. 

### 2.5. Data Analysis

Data from multiple sources including interview transcripts, independent field notes, and participants’ medical records were triangulated for a more holistic understanding of study participants. Each transcript was additionally peer-reviewed by both primary interviewers retrospectively to ensure consensus of information between the interviewers. 

Thematic analysis was then conducted across all transcripts, and relevant data in relation to the study objectives were identified and extracted to be indexed into one or more codes. Analyses were conformed to inductive and deductive codes relating to participants’ (i) socio-demographic background, (ii) motivations for drinking and (iii) against drinking, (iv) rationale for the high utilisation of emergency services, and (v) perceptions and/or experiences of this utilisation. Emergent themes were subsequently revisited by a third independent reviewer, and differences and similarities between participants were explored. For the purpose of reporting the key findings and supporting quotations in the present report, the assigned subject codes unique to each participant were used in replacement of their actual names.

Study participation did not affect medical care rendered to the patients, although several participants had an extended ED stay to achieve a greater degree of sobriety. Enrolled participants were reimbursed with SGD$100 (approximately USD$72) supermarket vouchers—which could not be used for alcohol—at the end of the interview. 

Ethics approval for the study was secured from the National Healthcare Group Domain Specific Review Board (DSRB 2020/00118). Informed consent was obtained from each participant, including permission to audio-record the interview and publish the data after removing identifiers to ensure anonymity. Our study adhered to COREQ guidelines [17].

## 3. Results

Forty patients were referred for the study and a total of 20 participants were enrolled when data saturation was observed (Figure 1). Interviews lasted an average of 60 min (ranging from 19 to 122 min) in a single session. It was estimated to be at the fifteenth participant where the saturation point was reached.

### 3.1. Study Participant Characteristics

Demographic characteristics are shown in Table 1. While both male and female participants were eligible, all 20 participants recruited were male. They were mostly Indian, with the remaining coming from different ethnicities prevalent in Singapore. The majority of them were either divorced or single.

Many participants had psychiatric conditions and various cardiovascular risk factors, including either, or a combination of, diabetes, hypertension, hyperlipidaemia, and ischemic heart disease.

Most reported that their first drink was between the ages of 10 and 20 years old, with a general preference for beer among other available alcoholic drinks. Aside from alcohol dependency, a majority of them were also tobacco users with an early onset age.

Table 2 further detailed the participants’ socio-economic characteristics. Educational background differed greatly among the participants, although most had completed only primary school education. Some of them attained post-secondary education and even various pre-tertiary qualifications.

Most participants lived in rental apartments or self-owned public housing. Those living in a 1- to 2-room apartment accounted for the largest proportion of interviewed participants, and only one participant was reported to be homeless. Some lived alone, while others were revealed to be living with their friends or family.

The majority of the study participants were unemployed at the time of the interview. More than half of them were previously incarcerated for various criminal offences. A significant minority had at least one previous suicide attempt. 

### 3.2. Reasons for High Utilization of Emergency Health Services 

#### 3.2.1. Treatment for Alcohol-Related Health Conditions

The majority of the interviewed participants reported experiencing alcohol withdrawal symptoms or various health conditions aggravated by their chronic drinking problem. They deemed these conditions severe enough to warrant immediate medical intervention, such that they had no choice but to call for an ambulance.


*I felt dizzy. I started to have alcohol withdrawal seizures. If I withdraw (from alcohol), I go into seizure… sometimes if I don’t drink, I get alcohol fits. I collapse in [sic] the ground. (Participant 020)*



*It’s only like you can’t move… then they lift your leg (and) you feel pain. Sometimes you really cannot get up. I know I really (cannot), I cannot [move] that’s why I call (the) ambulance. (If) I’m able to at least go (by) myself to (the) hospital, then I don’t have to call right? (Participant 019)*


#### 3.2.2. Treatment for Existing Health Conditions Unrelated to Alcohol Use

Most study participants were also afflicted with a single or a combination of chronic medical conditions, which led them to utilise emergency services due to overwhelming pain.


*…Go down to the shop (to) buy things and I don’t know what happened. (Then) I’ll be in the ambulance…I lose [sic] consciousness… I asked the paramedic “what happened? I (had) fits?” She said, “No, no, no. You were just lying unconscious.” And I said “I never drink… yet.” They told me, “…you suffered a mild heart attack.” From my previous epilepsy, this was my…first attack. (Participant 013)*



*[Regarding calling of ambulance] About two times a month… because of my pain. For me, I got [sic] hernia. Before that (I) had this chest pain, always get this chest pain. I do…go on my own (to the hospital). (But) when the pain is too…can’t bear the pain, there’s no choice, where (do) you want to go? (Participant 017)*


#### 3.2.3. Alcohol Intoxication That Led to Intervention by Concerned Bystanders

Some study participants shared that concerned members of the public, upon seeing them unconscious, were the ones who called for emergency services. Bystanders had judged the need for ambulance services since they were unsure why the participants were unconscious. Study participants themselves admitted that their condition at that present instance did not require any form of emergency intervention.


*I collapse in [sic] the ground. People got [sic] call 995… It’s not that I called… There are occasions when (the) public call [sic]. (Participant 020)*



*Sometimes, I sit (on) the bench, below the (housing) block, (people) say I fainted. When I lie down, somebody call (the) ambulance. Actually, I’m sleeping! (Participant 001)*


#### 3.2.4. Preference for High-Quality Service and Care Levels Afforded by Emergency Services and Departments

In general, most study participants felt a sense of gratefulness for the emergency services rendered, and the warmth and genuine care shown by emergency personnel. They appreciated the emotional involvement of these staff who not only recognised them but were also aware of their chronic conditions from multiple occasions of attending to them. The efficiency of the emergency services, which could render immediate medical help, further encouraged participants to continue utilising them instead of turning to other non-emergency options.


*The ambulance that comes…sometimes it’s the same people. So they know me, know what’s wrong with me…before I explain to them. They spend some time with me, then they see (that) I’m okay. It’s not that we are well-known, they just remember- remember- (me). (Participant 018)*



*If I come by ambulance, excellent service. Very fast, everything (gets) done. If I come personally by taxi, or (with) my sister or brother, then I [sic] got to wait and wait and wait. But ambulance is a very fast service. (Once) they come (and) they bring you out of the ambulance, they put you on the bed already. (Participant 015)*


Interestingly, several study participants also expressed guilt for their constant utilisation of emergency services and repeated ED attendances. They felt disheartened at their use of EMS and going to the ED, because they perceived their medical condition(s) to not require the level of service care provided. Some felt liable for causing delays in treatment for other patients due to their unplanned ED attendances and subsequent hospital stay. Generally, they recognised that emergency services and resources were needed by those who were in critical condition or had acute conditions that required immediate medical attention.


*Actually, I don’t want to call. Because when you call, your number is already there (in the system). Every time (when) the same person (keeps) calling, (the paramedics) don’t even come (anymore], because it is pulling a fast one… (When I go to the A&E because of drinking) I feel a bit guilty because I think these emergency services shouldn’t be used on me. (Participant 003)*



*Maybe I feel that it’s no good [sic], I feel sorry every time because the more people like me come here too often, it will delay and reduce the chances for other people (to use the emergency services or department). (Participant 011)*



*Here, accident [sic] people come down, heart problem, kidney problem, sometimes throat problem, a lot of cases. Sometimes some doctors also see me only [sic] (and they went,) “You again!” (Did) they think (that) I (really) want to come to hospital? (Participant 010)*


### 3.3. Motivations for Drinking 

#### 3.3.1. To Cope with Personal Issues

Study participants commonly cited alcohol consumption as a coping mechanism for various stressors. Some participants resorted to drinking as a way to release the frustrations that they experienced. Alternatively, some drank as a form of temporary pain relief. 


*(The reason) I drank the 17 bottles (of beer) was because of frustration. (I was) wrongfully dismissed, wrongly accused. And that’s the reason why I get very angry, very irritated, very frustrated, and I just simply drink and drink and (then go to) sleep. (Participant 003)*



*When the pain (gets unbearable and) I cannot take it [sic], (it) means (that) the medicine never work. I take beer to subside the pain. (I take) about two cans of beer, only 8%. But I never bluff you, alcohol (actually) never works. (Participant 009)*


Others turned to alcohol to fill the void from their loneliness or drank simply to pass time and cope with boredom. For instance, Participant 010, who was currently divorced, mentioned that loneliness and boredom contributed to his perceived need for alcohol consumption.


*Lonely. You know you’re divorced. (There is) nobody with you, no friends. (I) Want to cut (down drinking)… (but) sometimes you lonely, you feel boring [sic]… you stay alone, (it’s) very difficult! (Participant 010)*



*You see, I need to escape because of the boredom. So I took [sic] on the drinks. (Participant 001)*


#### 3.3.2. To Cope with Symptoms Suggestive of Alcohol Dependence

While some of the study participants persistently returned to drinking to manage alcohol-related withdrawal symptoms, others attributed their chronic alcohol consumption to wanting to experience the intoxicating effects of drinking. These symptoms or consequences of use could be either physical or psychological in nature.

The heavy drinking lifestyle, especially during periods of boredom, might be a reflection of alcohol dependency in study participants. In general, a sense of helplessness was observed in those who had perceived a need to consume alcohol for the abovementioned reasons.


*I tried to give up (drinking) on my own, but because of the withdrawal symptoms, then I just… I chose to go back to drinking rather than to undergo the withdrawal symptoms. (It feels) terrible, I tell you. Vomiting, shaking, your hands will shiver, your legs will tremble, that kind of thing [sic]. (Participant 002)*



*After a certain period then [sic] you don’t drink, you feel like you’re missing something. (It) feels like withdrawal or what [sic], (although) I’d say it’s not withdrawal. I’d say it’s sort of (the) mind. (Participant 014)*



*To be at peace, relax, and everything. (Participant 019)*



*Last time (it) was (to) drink to socialise, to just hang out with the uncles and pass time. The difference is (that) today, I just want to drink and I want to get drunk. I need to get drunk. (Participant 005)*


#### 3.3.3. Encouraging Social Environments

**Ease of Obtaining Free Drinks**. The vast majority of study participants were unemployed and did not have a financial source to sustain their drinking lifestyle. Despite so, participants mentioned the ease of obtaining free drinks from people around them. Like others, Participant 005 shared that acquaintances were willing to cover the costs of alcohol for them at nearby coffee shops, attributing this ease to his likeable personality. 


*But I must also say, it’s not so much of an issue, ‘cause back then (when) I was going to the coffee shop, I’m quite a well-liked person (and) I can have free drinks. (Participant 005)*



*But I never buy the liquor. My friend is a liquor shop owner so he brought the liquors. (Pariticpant 013)*



*…Bumped into a friend in my area, he said, “Wah, long time no see.” And I said, “Ya.” So he said, “Come, I buy for you [sic] beer.” (Participant 020)*


**Encouragement from Others to Drink.** The existence of a socially reinforcing environment appeared to be a strong enabler in promoting and maintaining one’s drinking behaviours, independent of their financial situation. Interestingly, Participant 003 reflected that his spouse had encouraged him to drink at a family event despite efforts at alcohol abstinence for the past few months. 


*That time (when) I was trying to go off alcohol for a few months, at a Christmas party my wife saw [sic] (that) I (was) very poor thing [sic], so she asked me to have a can of beer. But I was the first one to pass out. (Participant 003)*


While some study participants, such as Participant 003, received explicit encouragement to consume alcohol, others experienced more subtle nudges or reinforcements that facilitated the continuation of drinking behaviours. Several participants revealed that people around them would not raise the issue of their alcohol use as long as they fulfilled their job requirements and/or remained non-disruptive to others and were not judgmental about their alcohol use. 


*…(I was) late half an hour (for work), but they (are) ok. The office (is) very good to me. I never create trouble, drink [sic] also never argue with them, so they (are) all very good to me. (Participant 006)*



*Father knows I drink. He knows (that when it was) working time I drink. (Although) they know but they never disturb me. (Not even) one day also (did) he say “you don’t come [sic] to this church.” He never say. (Participant 017)*


**Benefits from Drinking Alone.** The decision to drink in solitude was prominent in many study participants. On one hand, it helped them avoid people who might express their disapproval to participants for their alcohol use. On the other hand, it also prevented them from getting into confrontations with others if they become intoxicated. Both benefits could have reinforced the drinking of alcohol.


*Nobody disturb. I’m staying alone. (Participant 001)*



*I prefer to drink alone with myself [sic]. I couldn’t resist it I feel uncomfortable at home and I don’t want to go out then bring [sic] into trouble. (Participant 019)*


#### 3.3.4. Alcohol Consumption Habits Were Simply Not Considered to Be Problematic 

Several participants did not perceive a problem in their alcohol consumption patterns. In other words, either their drinking behaviours were not seen as an indication of addiction, or they did not view their alcohol consumption to be as bad or serious as other forms of addictions such as drugs. In this case, alcohol addiction in comparison to drug dependency was deemed to be the “lesser of two evils”.


*I’m not a drinker, (although) I have drank all types of drinks. But I’m not a big addiction [sic] you know. I’m a drinker, I ever drink Chinese wine (and) everything… All I (have) drank before. But I’m not addiction [sic] until shivering, (that every) early morning must [sic] drink. (Participant 006)*



*My goal is to cut down drinking, (but I need to) cut down (on) the drugs first. (Cutting down the) drug [sic] is very important. I can’t afford to go for 7 years (in prison) you know, next sentence will be 7 years, I cannot sit (in prison). So my goal is to give up drugs (first). Drink [sic] I can (easily) cut down. (Participant 005)*


### 3.4. Protective Factors against Drinking in Study Participants

#### Temporary Psychological Respite Conferred by Formal Interventions

Participation in alcohol rehabilitation programs at various addiction treatment centres in Singapore was perceived to have some degree of effectiveness in reducing problematic drinking in individuals. Among those who received a previous formal intervention, participants cited the conducive and supportive environment during treatment as helping to suppress the urge to use alcohol, alongside an amplified resolve to improve their lifestyle. However, they returned to drinking upon discharge from these programmes, highlighting a limited effect in promoting alcohol abstinence. 


*NAMS [National Addictions Management Service] (have) already discharged me. WE CARE [a community addiction recovery centre] (also) feels that I’m basically okay. Because at least, I don’t drink and I go there [to the recovery centre]. It really helped… helped me (to) become a more healthy [sic] person and (engage in) positive thinking and (to be) more aggressive (about wanting to change for the better). (Participant 011)*



*You see, it’s an addiction. After (discharging) from IMH [Institute of Mental Health], after upon discharge (for) about 2 months I stayed away from alcohol. And then subsequently I took on (alcohol again), because boredom sets in, you know. (Participant 002)*



*At the addiction centre [National Addictions Management Service], you cannot contact (anybody) outside. Visitors also cannot come and see you. But it’s like a Hotel 81 [a brand of budget hotel in Singapore], got pillow got everything [sic], every type of food (that) you want… Chinese food, Indian food, Malay food, everything they’ll give you. But (there is) no contact with the outside world for 2 weeks. It’s a very good thing they did [sic]. After 2 weeks I feel better. Down there, they give (you) such a kind of environment that you won’t feel like drinking also [sic]. You know, got [sic] every kind of games down there… so you forget about drinking. (Participant 009)*


While many study participants acknowledged the usefulness of these treatment programmes, many did not express the motivation to take part as a means of managing their alcohol use. They knew that they had to be genuinely interested to help themselves and be committed to the treatment process. However, most were unwilling to reach out to and depend on external formal organisations for help.


*I prefer to do it [quit drinking] alone… I think the programmes [formal interventions] are good, but personally I feel more comfortable helping myself rather than engaging external organisations. (Participant 004)*



*I personally feel that the interventions are helpful but I’m not improving because I cannot accept (that I have to stop drinking) and I cannot stop drinking. I think we addicts, we don’t care for ourselves and so we don’t take professional advice seriously. (Participant 003)*


### 3.5. Deterring Social Environments 

One’s social environment could be both an enabler of and deterrence from alcohol consumption. Consistent with their inclination to drink alone, the discouragement or objection from important people in their lives could lead study participants to refrain from drinking, albeit temporarily, in their presence. These people may include family and even non-family members. 


*My sisters don’t like me to drink. Every time [sic] I get angry with them, they blame it on my drinking. My roommate (also) discourages me, so I usually don’t drink when he is around. My pastor, church people, people at WE CARE [a community addiction recovery centre] and NAMS [National Addictions Management Service] don’t want me to drink. (Participant 003)*



*If I go out with certain people, I cannot drink [because they do not like drinking] means I don’t drink. Suppose (if) I follow with my church people, I don’t drink, I don’t smoke in front of them. Even (if) I want to smoke, (and) I’m a heavy smoker, (if) they (are) around me means I won’t smoke. (Participant 006)*


### 3.6. Perceived Ability to Meaningfully Contribute to the Society 

Employment and day-to-day activities offer participants the opportunity to spend time constructively and meaningfully. Consequently, lower alcohol consumption was reported because of the need to work or attend church. Instead of idling away the hours in boredom or being confined at home, participants appreciated the opportunity to be meaningfully engaged, and this in turn reduced the perceived need to turn to drinking to occupy time. Some also indicated adherence to a drinking schedule so as not to jeopardise their employment, which could additionally serve as a method to assist with the management of their problematic drinking habits.


*Small can, 330? So maybe… [drink] seven to eight drinks per day? Why I said five drinks is because some days I go to church, I won’t drink. (Participant 020)*



*Before (when) I had a job, I had about three (drinks a week). Right now I got no job, (I drink) almost on an everyday daily basis. Because I (have) got no job, (I have) nothing to do at home. So if I am gainfully employed… then Singapore recognises me as a contributor to the society, then I will feel ok. But right now I got nothing to do. So what you want me to do at home? Sit down only (and) watch TV? No, right? (Participant 002)*



*We can have an interval of three days (without drinking), then we (will) drink again. Normally (if we do drink) it’s on Fridays because Saturday we got no job [do not need to work]. Friday (and) Saturday, can. Sunday don’t (drink), because Monday (we) have to wake up early to go to work. So, cannot. (Participant 015)*


## 4. Discussion

To the best of our knowledge, this is the first Asian qualitative study on this patient population. Based on the grounded theory approach, our study uncovered key themes for the high utilization of emergency health services, as well as perpetuating and protective factors concerning ARFAs’ alcohol misuse. We found that our study participants shared certain key demographic characteristics and had similar reasons for ED/EMS use. 

### 4.1. Demographic Characteristics

All our study subjects were male with a preponderance of Indians. In a recently published database study [18], using comprehensive nationwide Singapore data, the proportion of male patients vastly outnumbered female patients presenting with alcohol-related diagnoses. In the same study, Indians contributed a much higher proportion of patients compared to the other major ethnic groups in Singapore. This could explain why our study sample was mainly Indian males. This highlights the potential existence of certain unique and possibly cultural factors associated with their drinking behaviours. This is also consistent with another Singapore study [19] and warrants further research.

Most of our study subjects had lower education levels and were unemployed. Local data pertaining to unemployment in this population are lacking. Previous studies [20,21] have shown that problem drinking is associated with losing a job and remaining unemployed. This finding suggests that they had lower socio-economic status. This is consistent with other similar populations [5,22]. Low income was not a deterrent to a drinking lifestyle for the study population. This is ascribed to their socially reinforcing environments where obtaining drinks from others, including from their relatives, acquaintances, and friends alike, was effortless. While seemingly harmless, these small acts of ‘generosity’ created more harm, further entrapping them in a vicious cycle of alcohol misuse. These individuals surrounding the participant are known as “enablers” and may include family and friends. They may not have a conscious intent to encourage alcohol misuse, but their actions may inadvertently encourage our study participants to continue drinking behaviours. Efforts should be directed to involve “enablers” in the alcohol reduction strategies for the study participant.

### 4.2. Reasons for ED/EMS Usage

The reasons for using ED/EMS services fell into ‘push’ and ‘pull’ factors. Push factors included participants seeking treatment for alcohol withdrawal and treatment of health conditions not directly related to alcohol use such as chronic pain. In addition, the perceived need for urgent medical attention in alcohol misuse patients for conditions other than acute intoxication is well documented [5,23]. 

The main pull factor was the perceived high level of service provided by emergency personnel. Most participants in our study group had basic accommodation and seemed less likely to turn to the ED for the purposes of lodging or solely for food. They understood that they would receive high-quality, expeditious care when utilising these services and preferred this approach compared to routine specialist clinic-based alcohol services. Another qualitative study [11] had shown similar findings where ARFA patients had extremely positive accounts of their prior ED attendances.

### 4.3. The Challenge in Managing Behaviour

Perpetuating factors for continued alcohol usage were varied. First, having been exposed to heavy and constant alcohol consumption patterns since youth, our subjects had developed the habit of turning to alcohol as a coping mechanism for varying stressors. Some stressors included physical pain from unresolved physical pain. Most participants recognised that these effects only provided temporary relief. Second, they drank to cope with physical withdrawal symptoms. Third, enabling social environments perpetuated alcohol usage. Lastly, several study participants did not consider their drinking to be problematic.

In keeping with the lack of insight and motivation to seek appropriate help, a UK study [11] suggested that ARFA patients lack the motivation to attend specialist outpatient alcohol-specific treatment. Therefore, obtaining entry into formal interventions such as a detoxification program at an addictions centre is difficult, despite some ARFAs recognising their benefits. However, even with the completion of such programs, recidivism was high, possibly contributed to by enabling social environments. 

A paradigm shift from traditional treatment approaches and expectations is indicated. Research evidence points to the benefits of interventions involving case management and conducted primarily in home and community settings. In a pilot randomised controlled [24], Drummond et al. demonstrated that Assertive Community Treatment (ACT), a model of care that involves assertively seeing patients in the community and case-working to assist them in addressing and seeking help for problems in medical, psycho-social, and alcohol-use domains, led to less unplanned healthcare use. In a similar intervention model on the ARFA population, Hughes et al. [25] also showed reductions in both ED attendance and hospitalisations. 

An interesting finding from our study was the tendency of concerned bystanders to activate emergency services upon encountering intoxicated and poorly responsive individuals. This may be explained by the fact that Singapore is a densely populated city-state, with public areas such as void decks (communal areas on the ground floor of public housing) and food courts experiencing high footfall. Such study participants with public drinking tendencies may also be vulnerable to exploitation and harm during intoxicated states. Working along a harm reduction approach through psycho-education on safe drinking behaviours and environments may be of benefit. 

Overall, the observation that there could be more perpetuating than protective factors for alcohol use indicates a challenging endeavour to treat problematic drinking in our study participants. Alleviating the utilisation of EMS and EDs could be a long-term process. Exploration for interventions beyond traditional treatment approaches and expectations for such patients is indicated.

## 5. Limitations

The study was a single-centre qualitative study with a small sample size. As such, findings were exploratory, hypothesis-generating, and may not be generalisable to other ARFA populations. Secondly, findings may be subjected to social desirability, given the sensitive nature of the interview questions asked. Thirdly, the nature of convenience sampling led to selection bias in the enrolment process, since only patients who were available and willing to participate during the recruitment period were recruited. In order to minimise this bias, attempts were made to enrol patients who were unable to consent to be interviewed in an earlier ED visit. Finally, the study depended on self-reported data, which could not be verified for accuracy. 

## 6. Conclusions

ARFAs may have similar demographics and perpetuating and protective factors for alcohol misuse. Emergency health services could provide sought-after services to ARFAs, resulting in high utilisation. Multipronged social and medical intervention could improve drinking behaviours and decrease overall ED/EMS utilisation. Findings from this study should be validated through a robust, prospective cohort study.

## Figures and Tables

**Figure 1 ijerph-19-10795-f001:**
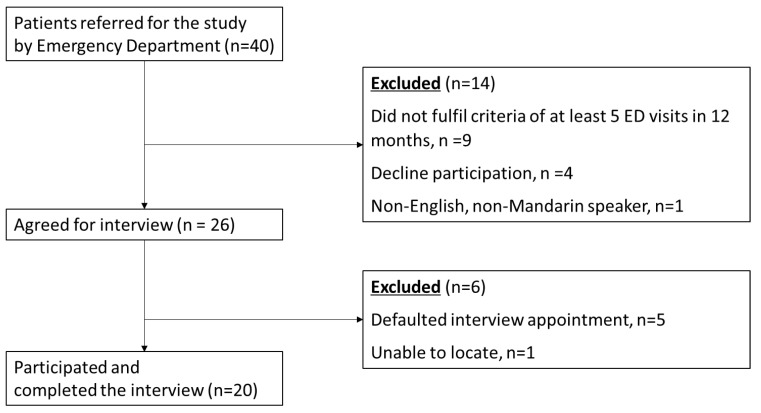
Flowchart of study enrolment.

**Table 1 ijerph-19-10795-t001:** Demographic characteristics.

Characteristic	Participants * (*n* = 20)
Age, y	55.6 (8.85)
**Gender**	
Male	20 (100)
**Ethnicity**	
Chinese	1 (5)
Indian	16 (80)
Malay	2 (10)
Sikh	1 (5)
**Religion**	
Agnostic	3 (15)
Buddhism	1 (5)
Christian	5 (25)
Hinduism	5 (25)
Islam	4 (20)
Sikhism	1 (5)
No religion	1 (5)
**Relationship status**	
Single	4 (20)
Married	2 (10)
Separated	1 (5)
Divorced	11 (55)
Widowed	1 (5)
Not available	1 (5)
**Medical History ^#^**	
Diabetes	4 (20)
Hypertension	11 (55)
Hyperlipidaemia	6 (30)
Ischemic heart disease	7 (35)
Any cardiovascular risk factor ^†^	15 (75)
Liver cirrhosis or previous hepatitis	6 (30)
Epilepsy	6 (30)
Psychiatric condition(s) ^^^	12 (60)
**Substance Use**	
**Alcohol dependence**	20 (100)
Onset of drinking between 10–20 years old	16 (80)
Preference to beer	15 (75)
**Smoker**	15 (75)
Age started smoking	16.7 (4.03)
Number of cigarettes daily	12.9 (10.28)
**Utilisation of Emergency Department**	
Number of Emergency Department visits in the last 12 months	15.4 (9.03)
Number of alcohol-related Emergency Department visits in the last 12 months	8.6 (6.72)

* Data are presented as mean (SD), or No. (%). ^#^ Medical history was retrieved from participants’ medical charts; ^†^ Diabetes, hypertension, hyperlipidaemia, or ischemic heart disease positive (any one), ^^^ Other than alcohol misuse disorder.

**Table 2 ijerph-19-10795-t002:** Socio-economic characteristics.

Characteristic	Participants (%) (*n* = 20)
**Highest Education**	
Primary school	7 (35)
Institute of Technical Education (Formal vocational training)	1 (5)
O Levels (General Certificate of Secondary Education) *	6 (30)
A Levels (General Certificate of Education Advanced Level) ^#^	1 (5)
Diploma ^%^	3 (15)
Did not answer	2 (10)
**Housing**	
1–2 room HDB ^^^ apartment	12 (60)
3–4 room HDB apartment	4 (20)
Condominium ^+^	1 (5)
Homeless	1 (5)
Not available	2 (10)
**Employment status**	
Full-time employment	2 (10)
Part-time employment	2 (10)
Retired	1 (5)
Unemployed	15 (75)
**Lives alone**	5 (25)
**Previous incarceration**	12 (60)
**Previous suicide attempt**	4 (20)

* Equivalent of US high school diploma, ^#^ Equivalent to US Advanced Placement course, ^%^ Equivalent of a US Associate degree, ^ Refers to public housing. 81% of citizens stay in public housing, ^+^ Privately owned or rented residential apartment.

## Data Availability

The data presented in this study are available on request from the corresponding author. The data are not publicly available due to privacy restrictions.

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
