# Peer review of "Why Are Some Male Alcohol Misuse Disorder Patients High Utilisers of Emergency Health Services? An Asian Qualitative Study"

_ijerph, 2022, doi:10.3390/ijerph191710795_

Round 1

Reviewer 1 Report (Previous Reviewer 2)

The study topic is important. However, there are major methodological concerns:

- only males were included in the study 

- 3/4 of respondents were unemployed

The sampling methods/recruitment methods are major concerns. Even if the analyzed phenomenon is more prevalent among males, this group is very limited. The study title as well as the other parts of the study should clearly define that this study is limited to the male population.

There is a lack of sufficient rationale for this study. Please justify in the Introduction section why this study (especially in Asia) is important?

Please improve the methods section to clarify the sampling methods. The Ethnicity of the participants is also limited to some groups that are not representative of the Singapore population. 

There is a limited novelty in this study.

Conclusions should be more precise.

Author Response

Reviewer 2 Report (New Reviewer)

Overall it is a very well-written paper discussing the reasons for repeated ED utilization among alcohol misuse patients.  One limitation, as the authors stated in the discussion section, was that the study has a relative small sample size.  The study would be much more informative if more participants from a broader demographic range could be recruited.  Besides, it would be great if the authors could discuss more regarding the relationship between previous psychiatric symptoms and problematic drinking-induced ED utilization. It would be great if few paragraphs discussing the addressed questions could be added to the discussion session.

Round 2

Reviewer 1 Report (Previous Reviewer 2)

The Authors provided responses, but most of the methodological concerns still exist. Only comment on the "rationale for this study" was addressed.

In the main text of the manuscript, questions on methodological concerns were not applied and revisions in the methods section are very small.

This study still has serious weaknesses resulting from (1) sampling methods; (2) study population; (3) characteristics of the population (unemployed males).

Moreover, the importance of this study for the international community is limited, as there are numerous limitations related to the methodology.

Author Response

This manuscript is a resubmission of an earlier submission. The following is a list of the peer review reports and author responses from that submission.

Round 1

Reviewer 1 Report

That's a highly important study, considering that people struggling with alcohol use disorder frequently have as their single source of health treatment emergency departments.

I have just a few comments that might improve the manuscript.

METHODS

- The study was conducted between Feb/July 2021, when Singapore was under phase 2 or 3 of the Covid pandemic. I was wondering if participants mentioned something about the impact of covid on their drug using habits and/or frequency of emergency department's visits.

Please include the equivalent in USD for SGD$100

RESULTS

- Table 1: Medical history was extracted from each patient medical chart, or is it self reported? If it's self-reported, please include a footnote, as those diagnosis could be inaccurate. 

Table 2: Is it possible to include some equivalent for their education? What does it mean to have "o level of education", is it completed high school? What does it mean to have a level education? Diploma is the same as college graduation? Please add footnotes, IJERPH has an international audience and many might not be familiar with the education system from Singapore.

Table 2: what is "HDB apartment"? "condominium" is a complex of townhouses? please add more information or at least a footnote- Is it possible to add at least the age of each participant after their quotation? This might help readers contextualize the idea...

DISCUSSION

This is NOT the "first Asian qualitative study on this patient population", there are many studies about this specific population. see, for instance (among others):

Liu Z et al. Nationwide Alcohol-related visits In Singapore's Emergency departments (NAISE): A retrospective population-level study from 2007 to 2016. Drug Alcohol Rev. 2022 Apr 19. doi: 10.1111/dar.13472. 

The authors should base their conclusions and recommendations on the study methods and findings. For instance, a small qualitative study cannot highlight "protective factors", it could instead suggest some areas to be further evaluated with more robust methods.

Although case-management has been highlighted by many studies as a successful strategy to manage alcohol use disorder, this specific study and its findings are not sufficient to corroborate this idea. Actually most selection quotes reinforced that those 20 patients felt welcomed in the ED and might continue utilizing this service as their main source of healthcare in the near future...

CONCLUSION

I would suggest that authors refer to "study participants" instead of "ARFAs", since those 20 study participants might not be representative of the entire population of patients struggling with alcohol use disorder that look for care in this specific hospital. They are certainly not representative of the entire population of patients with alcohol use disorder looking for care in ED all over Singapore - please be careful with over generalizations.

Thanks for including the study limitations!

Reviewer 2 Report

This study is of limited interest and has several methodological concerns.

  1. The Abstract section should be more informative. There is a lack of clearly defined methodology and results in the current version of the abstract
  2. The sample size is unclear. There is a lack of scientific-based justification for this sample
  3. The sampling methods are unclear. This kind of convenience sample may lead to the risk of bias.
  4. Please clearly define measures (e.g., attach a blank version of the interview list)
  5. Detailed characteristics of such a small sample size may be biased.
  6. There is a limited novelty in the findings

Round 2

Reviewer 2 Report

The Authors provided minor changes in the manuscript. Most of the questions were not addressed based on scientific standards. There are still methodological concerns regarding the sample size, sampling, etc. 
In the reviewer's opinion, this study is of limited international interest and should be submitted to some local journal.